# Development of a Method for Soil Tilth Quality Evaluation from Crumbling Roller Baskets Using Deep Machine Learning Models

**DOI:** 10.3390/s24113379

**Published:** 2024-05-24

**Authors:** Mehari Z. Tekeste, Junxian Guo, Desale Habtezgi, Jia-Hao He, Marcin Waz

**Affiliations:** 1Department of Agricultural and Biosystems Engineering, Iowa State University, Ames, IA 50010, USA; howardh@iastate.edu; 2College of Mechanical and Electrical Engineering, Xinjiang Agriculture University, Urumqi 830052, China; junxianguo@163.com; 3Department of Mathematical Sciences, DePaul University, SAC 504, Chicago, IL 60604, USA; dhabtzgh@depaul.edu (D.H.); mwaz@depaul.edu (M.W.)

**Keywords:** soil tilth, digital tillage, machine learning models, LiDAR soil profile

## Abstract

A combination tillage with disks, rippers, and roller baskets allows the loosening of compacted soils and the crumbling of soil clods. Statistical methods for evaluating the soil tilth quality of combination tillage are limited. Light Detection and Ranging (LiDAR) data and machine learning models (Random Forest (RF), Support Vector Machine (SVM), and Neural Network (NN)) are proposed to investigate roller basket pressure settings on soil tilth quality. Soil profiles were measured using LiDAR (stop and go and on-the-go) and RGB visual images from a Completely Randomized Design (CRD) tillage experiment on clay loam soil with treatments of roller basket down, roller basket up, and no-till in three replicates. Utilizing RF, SVM, and NN methods on the LiDAR data set identified median, mean, maximum, and standard deviation as the top features of importance variables that were statistically affected by the roller settings. Applying multivariate discriminatory analysis on the four statistical measures, three soil tilth classes were predicted with mean prediction rates of 77% (Roller-basket down), 64% (Roller-basket up), and 90% (No till). The LiDAR data analytics-inspired soil tilth classes correlated well with the RGB image discriminatory analysis. Soil tilth machine learning models were shown to be successful in classifying soil tilth with regard to onboard operator pressure control settings on the roller basket of the combination tillage implement.

## 1. Introduction

Humans have utilized wooden or steel tillage implements for centuries to plow and cultivate soil for planting and producing crops on farms [1]. Agronomic tillage studies that investigate soil engaging tools (GETs) on the soil–crop–climate environment have helped in selecting appropriate tillage depth, tillage frequency, and tool geometry design, which have resulted in reduced deep tillage energy and improved economical crop yields [2,3,4]. In the corn–soybean growing regions of the USA, it is becoming increasingly common to use an integral combination of primary and secondary tillage implements for soil tillage management. Combination tillage typically consists of shanks such as chisels or subsoilers for shattering compacted soils, along with secondary tillage implements like disk harrows and rotary baskets that are integral to the primary tillage implements (ASAE S414.2, 2018 [5]). After the combination tillage implement passes, shallow secondary tillage operations can be applied [4], which are intended to create a seedbed environment for establishing the crop during germination and emergence to achieve maximum crop yield potential. Depending on soil conditions (soil type, soil water content, and crop residue) during tillage, the design of tillage tools, tool depth, the speed of tillage operation, and integral tillage implement arrangements in the combination tillage implements, soil tilth quality is often difficult to qualitatively or quantitatively assess [4,6,7]. Various soil tilth assessment techniques have been studied. Colvin et al. (1984) [6] attempted to quantify tillage quality for soil conservation practices using four field soil and tool parameters: (1) soil height in row and inter-row zones, (2) soil roughness, (3) percent of crop residue, and (4) tillage depth. Tillage practices that leave large amounts of plant residue on the surface and improve surface water storage attributes receive a higher ranking as better soil conservation practices. Visual soil evaluation scoring, as reported by Ball et al. (2017) [8], may serve as a useful tool for assessing soil structure to investigate the effects of soil management practices from tillage and field machine tracking-induced soil compaction. Karlen et al. (1990) [7] reported historical studies on soil tilth, impacted by short-term or long-term effects of tillage systems, fertilization, crop rotation, and machine systems for managing soil resources for crop production and the environment. With the intent of developing a quantitative soil tilth quality index, Singh et al. (1992) [9] determined the soil tilth index as a multiplicative combination of coefficients rated in scales (0 to 1) of soil bulk density, soil cone index, soil organic matter, aggregate size distribution, and soil lower plasticity limit, where the coefficient of one refers to a soil environment that is favorable for root and plant growth and the coefficient of zero is assigned to soil conditions that are unusable for plants. The study reported a positive correlation of the soil tilth index to corn and soybean yields from various cropping systems (continuous corn and corn–soybean rotations) on fields that received tillage, with a coefficient of determination ranging from 0.15 to 0.86. As reported in Singh et al. (1992) [9], the measurement of the parameters to quantify the soil tilth index after tillage (disking and field cultivation) had values statistically significantly different from the soil tilth index measured before spring tillage. Separating soil tilth input data associated with real-time tillage tools and long-term cropping systems could improve the robustness of the quantified soil tilth index proposed by Singh et al. (1992) [9]. Earlier studies had implemented soil aggregation measurements as methods to classify tillage implement effects on tilled soils [10,11]. The soil tilth classification process using soil aggregate data after mechanical sieving is often time-consuming. Recently, new digital data collection methods with 2D or 3D RGB, laser technologies, and LiDAR have offered relatively quick data collection for tillage applications [12] and soil erosion management [13]. One of the knowledge gaps is to support the automation of setting combination of the primary and secondary tillage equipment on-the-go from the operator station with on-board soil tilth assessment. Developing soil tilth assessment for the real-time adjustment of combination tillage equipment during primary tillage applications could support tillage soil management for the seedbed environment. The increase in spring precipitation creates a soil condition that increases the likelihood of it being less than favorable for the efficient emergence of crops because of surface crusting and smearing at the seedbed trench. Soil management in fall can leave a loosened soil profile with residue mixed with the soil, reduce the need for disk tillage in spring on relatively wet soil conditions, and allow for better seedbed tillage management that could create an optimum soil environment for better seed-to-soil contact, plant emergence, and early plant growth.

Near-proximity soil clod size distribution or topsoil surface profile mapping using high-density data from Light Detection and Ranging (LiDAR) technology and algorithms to classify the soil tilth can have the potential to optimize the settings of tillage tools on seedbed tillage implement. Machine learning techniques based on LiDAR data can be developed and combined with available soil information databases, such as the one related to soil tilth parameters [9]. Machine learning model-trained soil tilth can then be integrated into tillage equipment to support better soil management decisions by design engineers or growers. The overall goal of this study is to develop machine learning methods with the potential to be used for adjusting the roller basket tillage finisher tools that are found as the last tillage implement interacting with soils from a combination tillage implement.

The main objectives of this study are to (1) investigate appropriate statistical measures using machine learning from the LiDAR tillage profile data set, (2) quantify the effect of roller basket hydraulic-controlled down forces settings on soil tilth quality, and (3) investigate machine-learning-trained classification of soil tilth differences for two roller basket settings and no-till.

## 2. Materials and Methods

### 2.1. Experimental Site Description

This experiment was conducted at the Agricultural Engineering and Agronomy (AEA) farm at Iowa State University (ISU) in Boone, IA, USA (latitude 42°01′ N, longitude 93°45′ W). Canisteo clay loam (*fine-loamy*, *mixed*, *superactive*, *calcareous*, *mesic typic endoaquolls*) was the dominant soil series at the experimental site according to the USDA soil survey (http://websoilsurvey.sc.egov.usda.gov/App/WebSoilSurvey.aspx, accessed on 1 January 2024). Table 1 shows a summary of soil physical and mechanical properties.

The tillage system experiment was applied using Completely Randomized Design (CRD) in three replications. Each experimental unit (100 m long by 3.6 m wide) received randomly assigned roller basket treatment levels of (1) roller-basket down, (2) roller-basket up, and (3) no-till. Each tillage treatment was replicated three times. Each experimental unit was divided into four sub-plots along the 100 m for sub-sampling data collection. The soil condition during the tillage application had a soil moisture content of 18.76% (d.b.) (Std of 1.04%; sample size (N) of nine) (Table 1). The soil moisture content during the tillage was 4.44% lower than the lower plastic limit (Ghorbani, 2019) [17]. John Deere (Moline, IL, USA) 8320R tractor pulling JD 2700 combination tillage equipment with 5-shank ripper (0.75 m shank spacing), disk, and roller basket was used to apply the tillage. The ripper and disk were set according to the fall tillage practice by the research farm operator of AEA at ISU. The experiment was performed by adjusting the set pressure in the tractor to control the roller basket down and up following the disk-ripper tools. No-tilled plots did not receive any of the ripper-disk-roller basket tillage, and measurement was taken on crop residue stable from the prior year’s soybean harvest. The roller-basket was adjusted by setting the rolling basket down-pressure in the tractor electro/hydraulic closed center systems to the maximum pressure setting. According to the implement operation manual, hydraulically lowering the rolling baskets is intended to create the basket in firm contact with the soil aggregate following the ripper-disk. For the roller-basket up, the basket down-pressure setting was put to the minimum pressure setting to create the least firming condition of the rolling basket in contact with the soil. Only the roller basket weight created the downforce on the soil for the roller-basket up configuration. Figure 1a,b show the tractor combination tillage implements and the experiment field plots after tillage treatment.

### 2.2. Data Collection Using Light Detection and Ranging (LiDAR) Unit

The Light Detection and Ranging (LiDAR) data were collected using SICK LMS511-10100, and SOPAS Engineering Tool (Minneapolis, MN, USA) acquisition software (Ver 2017). The LiDAR setting had an angular resolution of 0.5° and a sampling rate of 100 ms. Static and dynamic LIDAR data were collected on each subplot. The static data collection refers to the measurement taken when the truck had stopped before taking LiDAR reading (Figure 1c). For the dynamic LiDAR data collection, the truck with the fixture mounted at the back with SICK LiDAR LMS511-10100 was traveling at approximately 6 km/h (Figure 1d). The static LiDAR data collection can be classified as stop and go and the dynamic LiDAR approximates on-the-go LiDAR data collection. The scanning plane of the SICK LMS511-10100 was perpendicular to the forward direction of the truck. The fixture was built and mounted to the truck hitch with a fixed distance between the SICK LiDAR and black painted wooden plate. A fixed distance from the LiDAR bottom plane to the wooden top surface of 1.5 m (ho) was used to correct the LiDAR sensed range between the LiDAR and the soil surface.

### 2.3. Soil Surface Profile LiDAR Data Analysis

A program in MATLAB was developed to analyze the original LiDAR data to obtain a slice profile perpendicular to the longitudinal travel direction. Using the LiDAR slice profile data acquired at 0.5° increments, the height distribution of soil tilth quality parameters that quantify the tillage effects were determined. Each scan datum of the LIDAR profile had 377 points from −5° to 185° at the interval of 0.5°. Both ends of the original LIDAR data were removed before subsequent analysis was performed as they were not associated with the wooden plate and the soil surface. The analysis procedure included (1) correcting and obtaining the actual height from the rotation center of the LiDAR to the sample points of the soil surface, then (2) obtaining the reference vertical height to the black-painted wooden plate, and (3) analyzing the distribution parameters of the reference height to the black painted two wooden plates. For eliminating the disturbance from points out of two reference plates mounted to the right and left side of the frame, only the data set of 89 points from the 377 points referring to the data between the two reference wooden plates was used. The LiDAR data within the soil box without points from the two references were used to quantify the height distribution from the no-till and soil surface after tillage for the two roller settings (roller basket down and roller basket up). Detailed data analysis steps to generate the soil height profiles are explained as follows.

Step 1: True value of height hc from the rotation center of the LiDAR to the point pi of the soil is corrected by Equation (1)
(1)$$h_{c}=lcos(\alpha )\;$$

*h_c_* is the true value of the height of the point *p_i_* where *i* refers to the number of measurements points of the soil surface; l is the measured true value of height from the rotation center of the LIDAR to the point p, and α is the angle between the line of scanning points to the rotation center of the LIDAR and a vertical plane.

Step 2: The average height of the reference plate to obtain the reference height to the wooden plates is calculated using Equation (2)
(2)Ra=∑i=1nRin n=7

*R_a_* is the average height of the reference plate, and *R_i_* is the height of the point *i* of the reference plate. There are seven data points (*n* = 7) on each side of the reference plate (Wood width of 150 mm).

Step 3: The reference height h_r_ from the rotation center of the LIDAR to the point p_i_ of the soil surface is computed using Equation (3)
*h_r_* = *h_c_* − *Ra*(3)
where *h_r_* is the reference height from the rotation center of the LIDAR to the point *p_i_* on the soil surface. Within the measurement span width of the LiDAR, all the soil surface was tilled for the roller-basket down and the roller-basket up settings.

Statistics of the sampled LiDAR height data of *h_r_* from the soil surface swath width between the left and right wooden plates were calculated in MATLAB Statistics Toolbox (Matworks, R2023) as (1) maximum height value (*h_r_* Max), (2) minimum height value (*h_r_* Minimum), (3) mean height value (*h_r_* Mean), (4) standard deviation height value (*h_r_* Std), (5) median height value (*h_r_* Median), (6) mode height value (*h_r_* Mode), (7) skewness, (7) kurtosis of the data distribution, and (8) roughness coefficient (*g_rc_*). The roughness coefficient was calculated using Equation (4). The roughness coefficient (*g_rc_*) was included based on the maximum range to the mean height data after Tonietto et al. (2019) [18].
(*g_rc_* = MAX(*h_r_*_1_, *h_r_*_2_, …, *h_ri_*) − MIN(*h_r_*_1_, *h_r_*_2_, …, *h_ri_*))/MEAN(*h_r_*_1_, *h_r_*_2_, …, *h_ri_*)  *i* is the total number of point(4)

The roughness coefficient (*g_rc_*) quantifies the soil surface finish by comparing the relative percent of range (maximum–minimum) to the mean of the measured heights.

### 2.4. Statistical Analysis Using Machine Learning Techniques

Creating a finished soil tilth quality, for instance, roughness in field conditions is a result of soil machine operation variables, including soil type, soil moisture condition, crop residue percent and composition, and machine operator variables such as tillage tool design, dynamic responses of the hydraulic cylinder relative motion to soil conditions, and operator implement adjustments. Currently, machine operators set the down-force on the basket in combination with tillage by visually observing the soil’s relative roughness with limited data-analytics supporting methods. A robustness statistic variable is not available to numerically quantify the soil roughness as a function of the multiple parameters. Machine learning statistical methods offer potential techniques for learning such large LiDAR data sets from tillage operations in field soil conditions for analyzing appropriate statistical measures. Machine learning analytics and classified soil tilth classes are for operators to adjust the tool’s control; for example, the roller-basket pressure can be embedded into the tractor’s on-board display.

A machine learning statistical method according to ensemble methods has the potential to determine an appropriate statistic measure that quantifies soil tilth and investigates the effect of roller basket configuration on soil roughness. Ensemble methods are machine learning algorithms that use multiple learning techniques to obtain better predictive performance than could be obtained from any of the conventional statistical methods alone. Rokach (2010) [19] stated that ensemble machine learning methods can improve the prediction performance of variables that exhibit effects from multiple causes. The ensemble machine learning algorithms have been recognized as robustness methods used in non-parametric statistics and Artificial Intelligence (AI) applications. The ensemble methods evaluated in this study are Random Forest (RF), Support Vector Machines (SVM) and Neural Networks (NN). The RF, SVM, and NN models were selected due to their ability to identify the importance features and address multiclass problems. Our primary aim was to assess the efficiency of diverse parameters using these models, which are readily accessible through **Python** libraries such as **NumPy**, **SciPy**, and **sci-kit-learn**. R **ggplot2** and **tidyverse** packages were used for data analysis and visualization. The **coin** R package was employed for nonparametric analysis to compare roller setting treatments.

#### 2.4.1. RF Machine Learning Method

The RF is one of the most effective machine learning models for predictive analytics [20]. This type of additive model makes predictions by combining decisions from a sequence of base models (Figure 2a). The basis for the RF classifier is the decision tree, which uses a series of if-else conditions to classify observations like a flow chart. The diagram (Figure 2a) shows an example of a decision tree to predict which of two brands of a product a consumer will purchase based on an index measuring their loyalty to Brand A and the price difference between Brand A and Brand B. Consumers who are highly loyal to Brand A are predicted to purchase Brand A, while other consumers are predicted to purchase Brand A only if the price of Brand A is less than seven cents higher than that of Brand B. The RF tree is built using a process known as recursive binary splitting from the statistical measures estimated from the LiDAR data sets for the three tillage treatments. At each step, the algorithm splits observations into two groups, or nodes, using a cutoff value based on a single variable and trying to make an optimal split in terms of node purity.

#### 2.4.2. SVM Machine Learning Method

The SVM method was originally developed by Vapnik in the 1990s with a strong theoretical foundation in statistics and mathematics [21]. Wu and Kumar (2009) [21] stated that SVM is among the most robust and accurate methods in all well-known data mining algorithms. SVM allows for the misclassification of some training data, which allows us to fit a decision boundary even in cases where the points can not be perfectly separated and decrease the variance of the resulting predictions so that we do not see such dramatic changes in response to small changes in the training data. The goal is to allow some misclassification on the training data to obtain better performance on the testing data. The algorithm and the analysis procedure were implemented in the open-source software R (R version 4.1.2) and Python (Python 3.7.0). 

#### 2.4.3. NN Machine Learning Method

Neural networks are a broad class of machine learning algorithms, with many options for the modeler to customize. Indeed, some classical techniques such as linear regression can be formulated as a special case of NN models. The NN links the predictors (referred to as the input layer) to the response through one or more hidden layers, which can be envisioned as representing latent variables. The latent variables are not explicitly entered into the model but are nonetheless represented through linear combinations of the input variables. A set of weights defines how the linear combination is formed and is learned from the data. An activation function is applied to the linear combination that is sent to the next layer. In analogue with the brain, the elements of each layer are often referred to as neurons and the connections between layers as synapses. The activation function is then seen as controlling when and how each neuron fires in response to the inputs it receives. The figure (Figure 2b) below shows the structure of a typical NN with inputs feeding into hidden layers, which ultimately feed into an output layer.

### 2.5. VI Soil Surface Machine Learning

The VI method is applied using image extraction information and an RF algorithm to classify soil images from the three roller settings. The images were captured at approximately the same height as the LiDAR unit viewing perpendicular to the soil surface. Image input features by the roller settings were extracted through the following procedure: (i) all visible images (three replicates of the three roller settings) were cropped to a size of 65 × 65 pixels. Feature extraction from the RGB for each image was performed using the Visual Geometry Group (VGG10 model) to obtain 15 features per each RGB soil surface image using a similar image classification technique to that used by Simonyan and Zisserman (2014) [22]. For each RGB visible image, statistical descriptors such as mean, standard deviation, skewness, and kurtosis were calculated and combined with the features derived from the VGG10 model. As soil colors after being freshly tilled are affected by the level of soil moisture, for further enhancing the image classification and VGG10 model related to the soil tilth after the three roller settings, further image classification was performed by adjusting to the image of the soil condition at the lower plastic limit. The P index, obtained by comparing mean squared errors between visible images and soil images at the plastic limit, was incorporated with the above-mentioned features extraction techniques. The RF algorithm was then implemented on the resulting feature set (20 features per image) as input for the classification of the RGB visible image of soil surface using discriminatory regression analysis of the RGB estimated multivariate variables [23] to the three roller setting treatments.

Figure 3 shows the overall schematic of the block diagram for collecting LiDAR data after tillage, processing the LiDAR data set, analyzing it, and training machine models with features important for classifying the LiDAR trained to the three tillage classes.

## 3. Results

Soil moisture for the topsoil (0–150 mm) showed that the soil had a moisture content of 18.76% (d.b.), with a standard deviation of 1.04% and a coefficient of variation (CV) of 4.70%. The relatively low CV value indicates that the initial soil moisture condition of the experimental site was similar during tillage. The field soil moisture during tillage was 79% of the soil moisture content at plastic limit, and 61% of the field capacity soil moisture content of caniseo clay loam soil, as reported by Iqbal et al. (1998) [24]. Tilling at soil moisture at 79% of the lower plastic limit is expected to create larger soil clods as the soil moisture level approaches the wet workability limit, which is a soil moisture content close to the plastic limit and it is equivalent to soil water potential values (pF) of 1.9 (loam soil) and 2.2 (clayey soil) according to Hoogmoed et al. (2003) [25].

Figure 4 shows an example of a LiDAR slice profile of soil roughness acquired via static LiDAR data set after MATLAB profile analysis and 2D visual soil surface images for the no-till, roller basket up, and roller basket down settings. The LiDAR processed data and the visual images from the no-till without the combination tillage show that no soil clods were created. A comparison of the two roller basket settings after disk-ripping (roller basket up and roller basket down) shows that the basket rolling with only gravity external load (roller baskets up) created relatively shallower and spiked soil height distribution as compared to the roller basket down (setting with maximum cylinder pressure exerted on disk-ripped soil). Visual images also demonstrated the variations of soil height profile similar to the LiDAR processed profile among the three roller setting treatment levels. 

### 3.1. Evaluating the Fitness of the Three-Machine Learning Method

To properly evaluate the performance of each of the machine learning models, data of each LiDAR statistical measure (maximum height value (*h_r_* Max), minimum height value (*h_r_* Minimum), mean height value (*h_r_* Mean), standard deviation height value (*h_r_* Std), median height value (*h_r_* Median), mode height value (*h_r_* Mode), skewness, kurtosis of the data distribution, and roughness coefficient (*g_rc_*)) were obtained. The training– validation split employed a random split with an 80% training set and 20% test set using traintestsplit from the Python sklearn package. These percentages, 80% for the training set and 20% for the test set, were within the range proposed by Gholamy et al. (2018) [26]. Classification accuracy and feature importance values were calculated for each of the three machine learning algorithms. Accuracy is the fraction of predictions of the machine learning classifier to correctly identify the statistical measure to the corresponding data from the roller setting during the experiment. Feature importance scores can highlight which of the parameters are most or least relevant to identify the differences in the status of the roller; their values range from 0 to 1 [27]. The SVM, NN, and RF models from the static LiDAR data set averaged by the tillage treatments had measures of accuracy of 0.97, 0.96, and 0.98, respectively. Using the dynamic LiDAR dataset averaged by the tillage treatments, the SVM, NN, and RF measuring accuracy were 0.82, 0.80, and 0.81, respectively. The performance of the three classifiers, as reported by the measure of accuracy ranging from 0.81 to 0.98, can be considered accurate, with relatively higher accuracy from the static LiDAR data set than the accuracy from the dynamic LiDAR data set. The feature importance of the three machine learning models in classifying the statistical variables (maximum, minimum, mean, median, mode, standard deviation, kurtosis, skewness, and roughness coefficient) of the LiDAR soil profile after the three treatments (basket up, basket down and no-till) are shown for the static LiDAR data set (Table 2) and dynamic LiDAR data set (Table 3).

Ranked by the magnitude of the feature of importance from the static LiDAR data set (Table 2), SVM and NN selected median, mean, maximum, and standard deviation as top-ranked statistical measures of the LiDAR-generated soil tilth quality by the three roller settings, whether the roller basket was up or roller basket was down or none (no-till). The RF selected median and median (Table 2) as the two top features of importance. The three machine learning models selected median and mean as the top essential features. The two machine learning models (SVM and NN) identified maximum and standard deviation from the static LiDAR data set as high features of importance. Still, RF identified them (maximum and standard deviation) as low features of importance. Considering two of the machine learning models (SVM and NN) identified the four statistical measures of median, mean, maximum, and standard deviation. Further investigation using these measures was considered to provide investigating the impact of each of these settings on soil tilth quality.

From the dynamic LiDAR data set and the relative ranking of the magnitude of feature importance (Table 3), SVM and NN selected median, mean, maximum and standard deviation, and RF selected median and mean as good measures of classifiers of the LIDAR generated soil tilth quality for the roller basket whether the basket was up or down or no-till.

The roughness coefficient, a parameter often used for quantifying finished top surface profile, was found to have lower feature importance as compared to standard deviation, a statistical spread descriptor, by the three models (SVM, NN, and RF) from the static and dynamic LiDAR data set. The extreme maximum from the highly variable LiDAR data set from the no-till with the crop residue standing from the prior year might have created variability to lower the classifying strength of the roughness coefficient. Observing the three machine learning models in selecting statistical measures based on feature of importance, evaluating the median, mean, maximum and standard deviation variables generated from both the static and dynamic LiDAR data, these were selected for further analysis as classifier statistical measures for analyzing the effect of the three-roller setting (basket up, down or no-till) on soil tilth quality.

#### Two-Tailed Tests, Permutation Test, Kruskal–Wallis and Analysis of Variance (ANOVA) for the High-Importance Feature Measures

A summary of statistical parameters is presented in Table 4. An analysis of variance (ANOVA) was used to compare the mean change for the difference between the tillage treatments of the roller-basket down, roller-basket up, and no-till. A nonparametric permutation test and Kruskal–Wallis were added to test the level of significance differences among the three roller settings (up, down, none) when median, maximum, mean, and standard deviation are used as measures to detect the differences in tillage treatment. The permutation, Kruskal–Wallis, and F-tests were performed using a 2-tailed setting, with an adopted significance level of *p* < 0.05. All statistical analyses were performed using R and Python packages. The roller settings were found to statistically (*p* < 0.05) affect the median, mean, maximum, and standard deviation measures estimated from the static data set using the permutation and Kruskal–Wallis tests (Table 4). Testing using permutation and Kruskal–Wallis from the dynamic LiDAR data set, except the standard deviation (*p* = 0.04), all the other three measures (median, mean and maximum) varied statistically significant by the roller setting (*p* < 0.05). Based on the ANOVA, the mean estimated values from the static and dynamic LiDAR datasets were found to be statistically different based on the roller settings (*p* < 0.05; F-test value of 1621 for the static LiDAR dataset and F-test value of 1776 for the dynamic LiDAR dataset).

Figure 5 shows the box plot of the median, mean, maximum, and standard deviation for the three roller settings (no-till (0), basket down (1) and basket up (2)) from the static and dynamic LiDAR data sets. The trends on the median, 25% quantile (Q2) and 75% quantile (Q3) show decreasing in the absolute soil height with the changing the settings from no-till, roller basket down, and roller basket up, notably from the median, mean and maximum measures. For the dynamic data, there are some outliers; the outliers were detected as values less than the lower fences (Q3 − 2IQR) and greater than the upper fences (Q3 + 2IQR) where Q3 − Q2 is the interquartile range. Q2 is the 25% quantile and Q3 is the 75% quantile.

As reported in Table 5, the mean values from the no-till LiDAR statistical measures of the median, mean, and maximum were found to be numerically (absolute) the highest, followed by the data (median, mean, and maximum) from roller basket down setting. The roller basket-up created the smallest (absolute values) for all the measures (median, mean, and maximum) of the static and dynamic LiDAR data sets. Assuming the no-till mean and median data represent soil surface profile before tillage, the differences in the median and mean no-till and the two roller settings (roller basket up and roller basket down), roller basket up, resulted in a difference of the median and mean by 34.64 mm and 31.27 mm, respectively from the static LiDAR data set. Median and mean differences no-till versus roller basket down were 27.45 mm and 25.42 mm, respectively. Changing the setting of the roller basket from up (roller basket weight) to exerting maximum cylinder pressure (roller basket down) on the disk-ripped soil resulted in reducing the median by 7.19 mm and mean by 5.85 mm. Analysis of the dynamic LiDAR data set measures and the effects of roller settings after disk-ripped soils as compared to the no-till, simulating before tillage conditions, maximum exerted cylinder pressure (roller basket down) decreased the median and mean values by 8.92 mm and 8.36 mm, respectively. The standard deviation values as a measure of the spread of the LiDAR soil heights from the three settings showed a similar trend as the other measures (median, mean, and maximum) from the static LiDAR data set (Table 5a). The differences in the standard deviation values from the static LiDAR and dynamic LiDAR data sets from the roller basket down and roller basket up were less than 0.5 mm (Table 5b). The coefficient of variation (COV) values (COV = standard deviation/mean) for each roller setting were higher from the dynamic LiDAR data set as compared to the COV from the static LiDAR data set.

### 3.2. Discriminatory Analysis

A discriminatory analysis was performed by implementing multivariate regression best fitting model based on estimated distances from each observation to each group’s multivariate mean (centroid) of the top-four feature importance statistical measures such as median, mean, maximum, and standard deviation. The discriminatory analysis then predicts the closest associations of the four statistical measures and their corresponding roller-setting classes, resulting in discriminatory scores. The discriminatory analysis was also done to visual image (VI) features taken from the three roller settings (basket up, basket down and no-till). The confusion matrix, showing the true labels from the LiDAR measured data set, and predicted labels after applying the discriminatory analysis are shown in Figure 6 for the VI, static LiDAR data set and dynamic LiDAR data set. Predicted labels (counts) versus truly labels (from the CRB tillage experiment measured and analyzed LiDAR) for no-till (predicted) versus no-till (true), basket up (predicted) versus basket up (true) and basket down (predicted) versus basket down (true) had counts higher as compared to the confused (wrongly mismatched) roller setting counts from the three data sets of VI, LiDAR static and LiDAR dynamic (for example, nine and four of the true no-till images were wrongly (confused) to be predicted as basket up and basket down, respectively from the VI data set (Figure 6a). Having maximum predicted counts for each of the true labels (counts) of no-till, basket up and basket down using image, and the four top-feature importance statistical measures (median, mean, maximum and standard deviation) of the LiDAR data sets (static and dynamic) suggests that the machine learning models techniques and LiDAR data can satisfactorily discriminate soil tilth by the roller settings of basket up (weight of the rollers) and basket down (maximum cylinder pressure exerted on the rollers) or no-tilled soil surfaces.

The prediction rate values for labelling no-till as no-till using the three methods (VI imaging, dynamic LIDAR and static LIDAR) were found to be the highest for each classification method (93% imaging, 91% dynamic LIDAR and 89% static LiDAR) (Table 6). Overall, the prediction rate of labelling basket up and basket down was found to be higher using the LiDAR data sets (both dynamic and static) as compared to the prediction rate of both basket settings from the visual imaging technique. The data suggests that using only visual images relatively closer to human experience classification was ineffective for quantifying the soil tilth quality from disk-ripper-roller combination tillage implementations. Classifying the soil tilth either roller basket down (highest pneumatic cylinder press) was correctly classified at prediction rates of 58% from the dynamic LiDAR data set and 89% from the static LiDAR data set. The relatively lower classification accuracy from the dynamic LiDAR data set could be due to the variation of keeping the cylinder pressure on the roller while moving the vehicle at 6 km/h. At the setting when the basket was up (applying the weights of roller baskets), the differences in the prediction rate using dynamic (70%) and static LiDAR (65%) were only 5%. Mislabeling of the pair of the two roller basket settings (down versus up, and up versus down) had a prediction rate approximately 30% among the three methods with smaller improvements when using static LiDAR data set.

## 4. Discussion

This study shows methods to quantify soil tilth caused by soil crumbling actions of two settings of a roller-basket, the last soil-engaging tool of a combination tillage implement of a disk-ripper, on clay loam soils. The data analytics using LiDAR soil profile, the visual image analysis, and machine learning modeling techniques numerically quantified the soil tilth assessment. Adjusted by setting the rolling basket pressure using the tractor electro/hydraulic closed center systems to the maximum pressure setting created soil tilth quality of shorter soil height and uniform across the basket rolled after disk-ripping than the soil height characteristics from the no-tilled surface, and crumbled by using only the weight of the roller basket. Implementing machine learning analytics with discriminatory soil tilth classes using the machine learning models on canisteo clay loam soil can enhance digital tillage artificial intelligence for manual or automated adjustment of a combination or individual soil-engaging components of the disk-ripper-roller basket soil management tillage system. The soil tilth classification may not necessarily replace the cultural practices of tillage operators; rather, the intelligence of deep-learning data analytics of the soil-to-tool interaction could enhance the efficiency of soil management for seed-bed preparation and enhance early crop growth.

Being able to collect LiDAR soil profiles dynamically (6 km/h) after disk-ripper-roller baskets at a speed close to the tillage implement speed (8 km/h) and demonstrating statistically significant differences from roller basket settings as the static LiDAR estimated soil roughness measures (median, mean, maximum and standard deviation) suggests that the LiDAR-data-set-trained machine learning models could be utilized for evaluating soil tilth during primary tillage operations. On Nicollet, fine sandy loam soils, closer soil series to the Canisteo soil series at the experiment site, Adam and Erbach (1992) [10] reported a mean weighted diameter (MWD) soil clod size of 22.8 mm after disk-harrow tillage at an initial mean soil moisture content of 21.65% (d.b.). It seems that the findings on the effects of roller basket setting after the disk-ripper as compared to the no-till on the difference of LiDAR median values between the roller basket down and no-till (27.45 mm) and between the roller basket up and no-till (34.64 mm) from the static LiDAR data set are consistent with the effect of disk-harrow tillage on laboratory-measured MWD of soil clods of 22.8 mm on Canisteo soils [10]. The higher median LiDAR estimated values after the disk-ripper-roller basket compared to MWD soil clod created by disk-harrow [10] could be due to relatively lesser soil crumbling by the roller baskets after deeper subsoiling (300 mm) on relatively wetter initial soil conditions (18.35%, d.b.). Assessing the soil tilth quality comparing roller crumbling after disk-ripping and no-till appeared to be similar to previous reports such as the Fanigliulo et al. (2020) [12] study to assess plowed, harrowed, and no-till soil using data from light Drone-Based 3D RGB. Implementing machine learning analytics with discriminatory soil tilth classes in this study on clay loam soil tilled using a combination tillage implement (disk-ripper-roller basket) demonstrates that digital tillage artificial intelligence (AI) is better than the traditional methods of time-consuming manual sieving methods [10] or visual operator qualitative soil finish assessment. The machine learning and discriminatory methods deployed in our study could also be implemented for other data collection techniques (such as light drown RGB 3D used by Fanigliulo et al., 2020 [12]) and instrumented tines for measuring aggregate size on-the-go proposed by Bogrekci and Godwin (2007) [11]. In dusty soil environments, which could occur during tillage operation on drier soil conditions, the ranging and light detection for the dynamic LiDAR data collection might be affected. Using static LiDAR collection could be a better technique if dusty soil conditions are created behind the tillage passes.

## 5. Conclusions

The following conclusions were drawn from this study for developing machine learning models to classify soil roughness estimated from LiDAR-trained statistical measures from three roller settings on clay loam soil.

Machine learning methods were successfully used to evaluate soil tillage quality soil profiles as a function of basket roller pressure settings of disk-ripper-roller baskets on clay loam soil. Three machine learning models (SVM, NN, and RF) were used to select statistical measures of median, mean, maximum, and standard deviation as the top four important features and varied statistically significant by the three roller settings.The roller basket down setting induced by applying the maximum hydraulic pressure exerted on disk-ripper soil created more uniform soil tilth characteristics, as quantified by median, mean, maximum, and standard deviation LiDAR soil profiles as compared to soil tilth created from roller basket up. The baseline reference of no-till (crop residue surfaces) without tillage was statistically different from the two soil profiles of the two basket roller settings.A comparison of LiDAR data collection using static (stop-go) and dynamic (on-the-go) methods appeared to show a similar classification of the tilled soil tilth after applying the machine learning models and discrimination analysis, with the data from the dynamic LiDAR data set creating higher standard deviation and outliers.The proposed machine learning analysis techniques from static and dynamic LiDAR soil data sets can further integrate digitalized soil tilth classes into an operator control display for manual or automated adjustment of roller basket settings from disk-ripper-roller basket tillage passes of combination tillage implements.Further research could be helpful in evaluating the proposed techniques for comparing tillage operations of various primary and secondary tillage implements and soil tilth classified trained using machine learning on crop growth and crop yield effects.

## Figures and Tables

**Figure 1 sensors-24-03379-f001:**
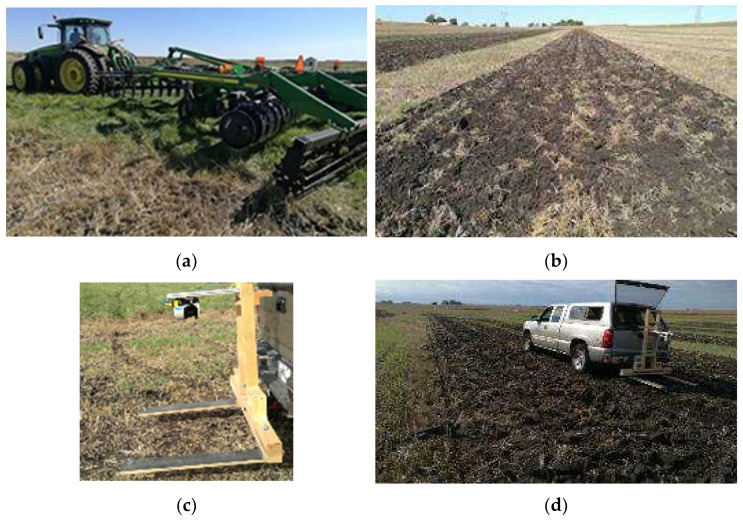
The experimental site where tillage tilth measurements were taken showing (**a**) John Deere 8320R tractor pulling JD 2700 combination tillage equipment. The configuration of the soil engaging tools in the tillage equipment had disk-ripper-disk and roller basket; (**b**) visual soil roughness conditions on the tilled and no tilled-tilled sampling zones from left to the right sections of the image, (**c**) side view of a SICK LiDAR measurement system mounting fixture to a truck hitch and wooden reference frame for static LiDAR data collection, and (**d**) side view of a SICK LiDAR measurement system mounting fixture while collecting dynamic LiDAR data set at 6 km/h truck forward velocity while taking SICK LiDAR data set collection.

**Figure 2 sensors-24-03379-f002:**
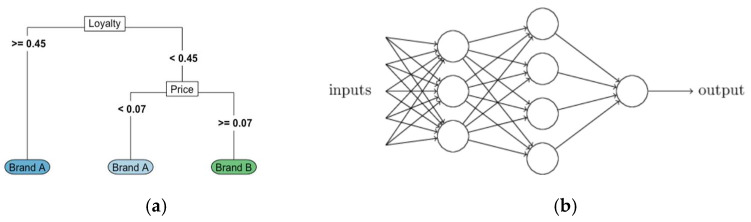
A simple tree machine learning schematic approach based on RF classifier [20] (**a**), and schematic NN layout (**b**).

**Figure 3 sensors-24-03379-f003:**
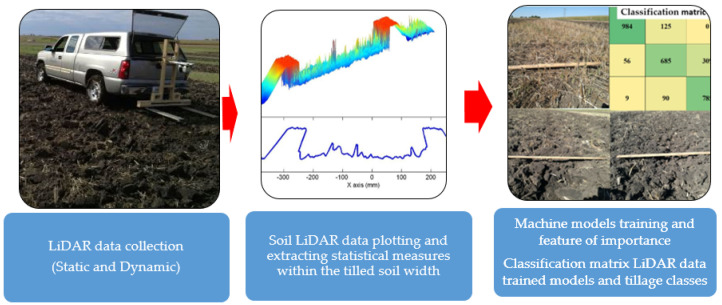
A schematic block diagram of the methods for collecting LiDAR data after tillage, LiDAR data set processing, analysis and machine model training with feature of importance for classification of the LiDAR trained to the three tillage classes.

**Figure 4 sensors-24-03379-f004:**
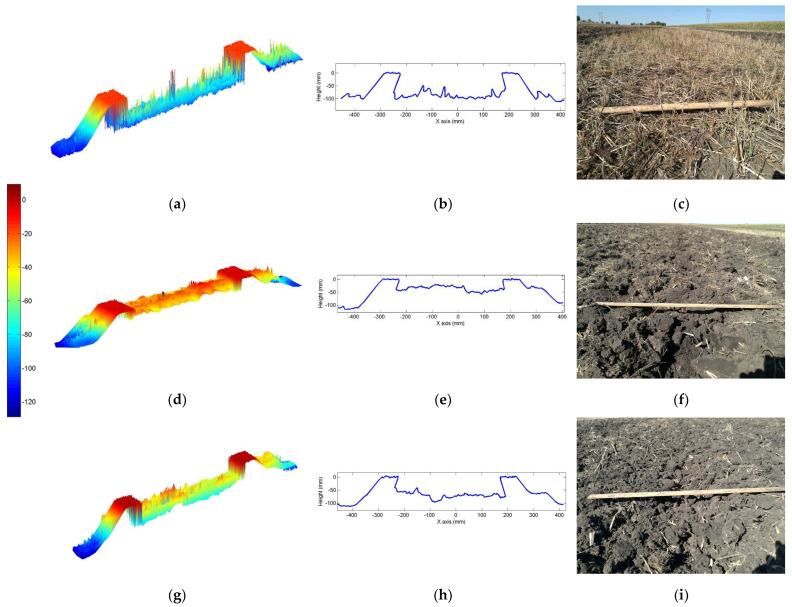
Sample of soil profile after MATLAB plotted profiles of static LiDAR data set, soil slices and field visual profile of soil from the no-till (**a**–**c**), roller basket up (**d**–**f**) and roller basket down (**g**–**i**).

**Figure 5 sensors-24-03379-f005:**
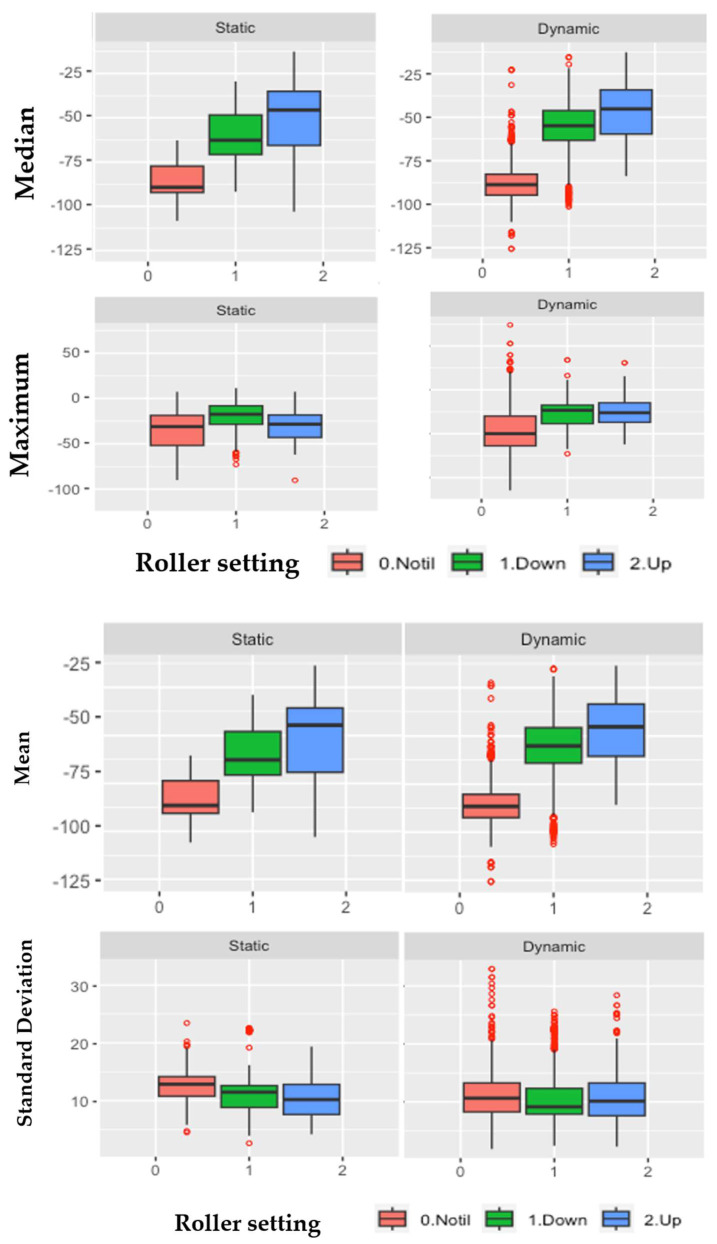
Box plot of median, mean, maximum, and standard deviation on the static and dynamic LiDAR data sets from the three roller settings (no-till, basket down, and basket up).

**Figure 6 sensors-24-03379-f006:**
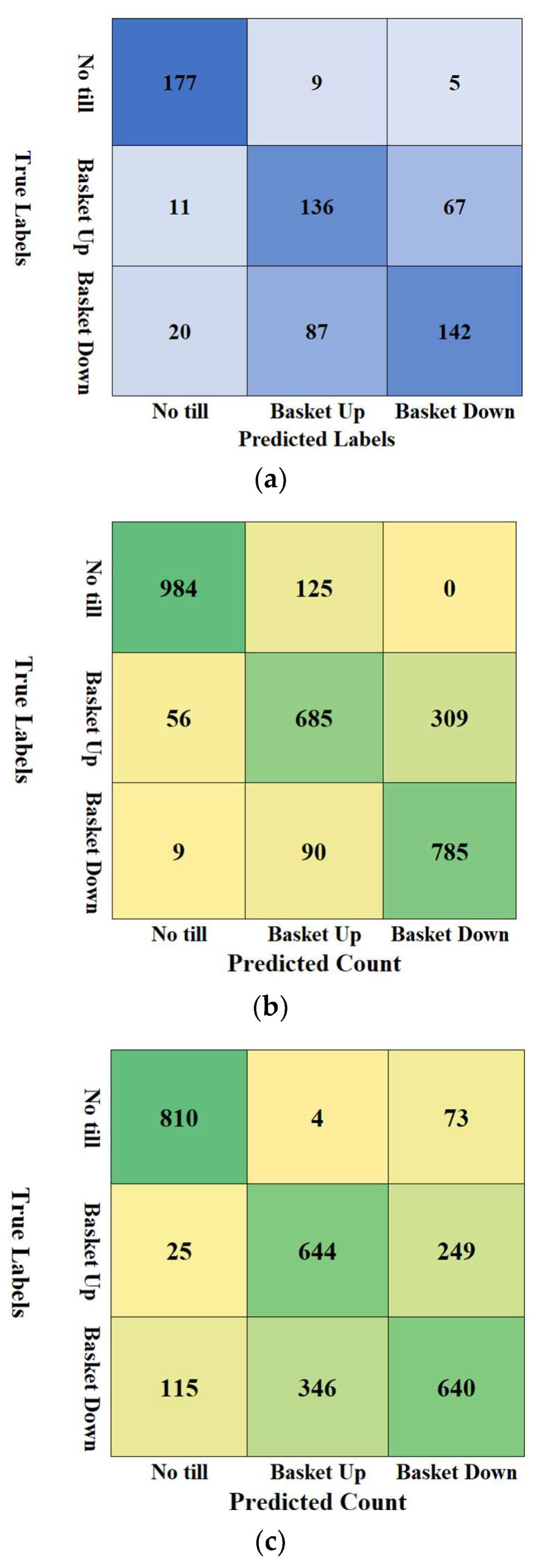
The confusion matrix from RGB visible imaging analysis (**a**), static LiDAR estimated measures of median, mean, maximum, and standard deviation (**b**), and dynamic LiDAR estimated measures of median, mean, maximum, and standard deviation (**c**).

**Table 1 sensors-24-03379-t001:** Descriptive statistics of soil physical and mechanical properties of the loam soil.

Percent Particle Size (%) ^1^ (Std)			Soil Moisture Content (%, d.b.) (Std)			Organic Matter ^2^ (%) (Std)	PL, W_p_ ^3^ (d.b., %)	LL, W_L_ ^3^ (d.b., %)
Sand	Silt	Clay	No Till	Roller Basket Up	Roller Basket Down			
27.5	43.3	29.2	18.68	18.32	18.06	4.7	23.2	31.9
(2.5)	(3.8)	(1.4)	(1.11)	(0.78)	(0.87)	(0.2)	(1.9)	(0.5)

^1^ According to USDA-Textural Soil Classification, sand fraction ranges from 2 mm to 0.05 mm, silt from 0.05 mm to 0.002 mm; and clay less than 0.002 mm (Gee and Bauder, 1986 [14]). ^2^ Soil organic matter was total soil organic carbon measured using “loss-on-ignition” dry combustion method (Nelson and Sommers, 1996 [15]). ^3^ PL, W_p_: water content in dry basis (%) at the lower plastic limit, and LL, W_L_: water content in dry basis (%) at liquid limit measured according to ASTM D4318 [16].

**Table 2 sensors-24-03379-t002:** Feature importances of the statistical measures from the static LiDAR data set from the three machine learning models of SVM, NN, and RF.

Parameter	SVM	NN	RF
	Feature Importance		
Median	0.231	0.228	0.224
Mean	0.137	0.199	0.219
Maximum	0.143	0.281	0.048
Standard deviation	0.227	0.277	0.071
Mode	0.06	0.094	0.083
Minimum	0.114	0.197	0.038
Kurtosis	0.128	0.18	0.092
Skewness	0.128	0.18	0.092
Roughness Coefficient	0.083	0.178	0.039

**Table 3 sensors-24-03379-t003:** Feature importances of the statistical measures from the dynamic LiDAR data set from the three machine learning models of SVM, NN, and RF.

Parameter	SVM	NN	RF
	Feature Importance		
Median	0.155	0.251	0.164
Mean	0.193	0.198	0.204
Maximum	0.095	0.158	0.089
Standard deviation	0.120	0.142	0.081
Mode	0.074	0.267	0.084
Minimum	0.021	0.025	0.108
Kurtosis	0.055	0.120	0.109
Skewness	0.182	0.167	0.108
Roughness coefficient	0.07	0.043	0.053

**Table 4 sensors-24-03379-t004:** Two-tailed tests for comparison of the three roller basket settings from the static LiDAR data set (a) and from dynamic LiDAR data set (b).

**Static LiDAR Data Set (a)**		
**Parameter**	**Permutation Test**	**Kruskal–Wallis**
Median	T = 40.7 (*p* < 0.0001)	Χ^2^ = 1746.2 (*p* < 0.0001)
Mean	T = 38.9 (*p* < 0.0001)	Χ^2^ = 1585.6 (*p* < 0.0001)
Maximum	T = 10.43 (*p* < 0.0001)	Χ^2^ = 117.17 (*p* < 0.0001)
Standard deviation	T = 13.25 (*p* < 0.0001)	Χ^2^ = 291.16 (*p* < 0.0001)
**Dynamic LiDAR Data Set (b)**		
**Parameter**	**Permutation Test**	**Kruskal–Wallis**
Median	T = 40 (*p* < 0.0001)	Χ^2^ = 1625.6 (*p* < 0.0001)
Mean	T = 38 (*p* < 0.0001)	Χ^2^ = 1551.2 (*p* < 0.0001)
Maximum	T = 22.6 (*p* < 0.0001)	Χ^2^ = 488.4 (*p* < 0.0001))
Standard deviation	T = 2.4291 (*p* = 0.04)	Χ^2^ = 18.025 (*p* = 0.0001)

**Table 5 sensors-24-03379-t005:** Mean and standard deviation of median, mean, maximum, and standard deviation for the three roller settings (No till, basket down and basket up) from the static LiDAR data set (a) and dynamic LiDAR data set (b).

**Static LiDAR Data Set (a)**				
	**Median (Std)** **(mm)**	**Mean (Std)** **(mm)**	**Maximum (Std)** **(mm)**	**Standard Deviation (Std)** **(mm)**
Basket down	−57.37 (12.62)	−57.14 (13.00)	−24.02 (17.14)	11.04 (3.97)
Basket up	−50.18 (15.52)	−51.29 (16.49)	−30.81 (15.05)	10.46 (3.19)
**Dynamic LiDAR Data Set (b)**				
	**Median (Std)** **(mm)**	**Mean (Std)** **(mm)**	**Maximum (Std)** **(mm)**	**Standard Deviation (Std)** **(mm)**
Basket down	−55.29 (14.42)	−55.22 (15.05)	−27.33 (15.10)	11.14 (5.27)
Basket up	−46.37 (15.29)	−46.86 (15.43)	−24.98 (15.34)	10.73 (3.94)
No till	−87.21 (12.57)	−85.31 (12.57)	−45.28 (25.50)	11.22 (4.43)

**Table 6 sensors-24-03379-t006:** Prediction rate using the discriminatory analysis, comparing the true labels and predicted labels for the three roller settings of (No-till, basket down and basket up) from visible imaging (a), static LiDAR data set (b) and dynamic LiDAR data set (c).

**Visible Imaging Analysis (a)**				
		**Predicted Labels**		
		**No till**	**Basket Up**	**Basket Down**
True Labels	No till	92.67%	4.71%	2.62%
	Basket Up	5.14%	63.55%	31.31%
	Basket Down	8.03%	34.94%	57.03%
**Static LiDAR Data Set (b)**				
		**Predicted Labels**		
		**No till**	**Basket Up**	**Basket Down**
True Labels	No till	88.73%	11.27%	0.00%
	Basket Up	5.33%	65.24%	29.43%
	Basket Down	1.02%	10.18%	88.80%
**Dynamic LiDAR Data Set (c)**				
		**Predicted Labels**		
		**No till**	**Basket Up**	**Basket Down**
True Labels	No till	91.32%	0.45%	8.23%
	Basket Up	2.72%	70.15%	27.12%
	Basket Down	10.45%	31.43%	58.13%

## Data Availability

The authors will make the raw data supporting this article’s conclusions available upon request.

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
