# Peer review of "Development of a Method for Soil Tilth Quality Evaluation from Crumbling Roller Baskets Using Deep Machine Learning Models"

_sensors, 2024, doi:10.3390/s24113379_

Round 1

Reviewer 1 Report

Comments and Suggestions for Authors

This work developed machine learning models to classify soil roughness estimated from LiDAR-trained statistical measures. However, the methodology employed lacks novelty, and there are notable deficiencies in the experimental design. I recommend the authors address the following points before resubmitting the manuscript:

Major comments:

1. The three machine learning models (i.e., SVM, random forest, Neural Network) are not novel. The mean prediction rates (77% (RBD), 64% (RBU), and 90% (NT)) are not high enough. I suggest exploring more advanced models.

2. As outlined in Section 2.3, each LIDAR profile consists of 89 valid points. However, the authors only utilize limited statistical information from the LIDAR data, failing to fully exploit the available profile points, thereby negatively impacting model performance.

3. The detailed parameter settings for each model are unclear. Given the significant impact of parameter values on prediction performance, it's crucial to specify parameters in Section 2.4. Additionally, including a parameter setting analysis would demonstrate how to determine optimal parameters for your applications.

Minor comments:

-The number of LIDAR profiles collected is not mentioned in Section 2. Please specify the quantity of LIDAR data used.

-Line 224: Please ensure the necessary reference is added at this point.

-Line 283-284: The figure depicting the SVM decision boundary is not provided as stated.

-Line 311-317: Clarification is needed regarding the implementation of image classification.

-Line 351: Consider using a 70% training and 30% testing split and justify the choice.

-In Figure 5, ensure consistency in the sizes of squares in the confusion matrix.

Author Response

Please see the attached document for responses from authors for reviewer-1

Reviewer 2 Report

Comments and Suggestions for Authors

The authors propose using a combination of tillage techniques involving a disk-ripper-roller basket to improve soil quality in compacted soils. They suggest using Light Detection and Ranging (LiDAR) data and machine learning models (such as Random Forest Trees, Support Vector Machines, and Neural Networks) to assess the impact of roller basket pressure settings on soil quality. They compared three treatments: roller-basket down (RBD), roller-basket up (RBU), and no-till (NT). 

In general, the paper is very well written and explained. Additionally, 3 different ML models for the proposed application are presented and compared. I only have a few comments on this paper.

-In the materials and methods part, you must start with a figure that describes the architecture of the proposed procedure or system in the form of a block diagram. For example, in this case, the stage of data acquisition, data analysis, pre-processing (if required), machine learning (data mining stage), and analysis of results is necessary.

-I recommend separating the discussion section from the results section. In the discussion section, the most important findings found during the development of this work should be detailed. On the other hand, in the conclusions, there should only be a summary of the work carried out and brief comments on the results of the paper.

- Please, Indicate the software and hardware used to train the machine learning models. Likewise, indicate the inference time of each ML model.

-In the discussion section, indicate possible problems of this architecture due to the environment and system used. For example, how does dust or dirt raised from the ground affect data acquisition?

-Please, Increase the size of Figure 4 to improve the readability of the text in the figure

If the authors make all the corrections I suggest for this article, it should be published without any problems.

Comments on the Quality of English Language

Minor editing of English language might be required

Author Response

Please find the attached document for responses from authors for reviewer-2

Reviewer 3 Report

Comments and Suggestions for Authors

1. The article demonstrates high classification accuracy using RF, SVM and NN models. However, the details of the validation process for these models are not fully covered. It is recommended to conduct cross-validation or use delayed sampling to evaluate the performance of models to ensure their generalizability and robustness to new data.

2. To improve understanding and decision making based on machine learning models, it is desirable to provide an interpretation of the results obtained. For example, feature importance in RF can help understand which aspects of LiDAR data are most significant in determining the quality of soil structure.

3. To strengthen confidence in the results of the article, it is necessary to describe in more detail the data collection procedure, including the calibration of LiDAR equipment and the accuracy of measurements. It is also important to ensure that experimental conditions (such as soil moisture and temperature conditions) are controlled and documented properly.

4. When analyzing data, it is advisable to use not only machine learning, but also traditional statistical methods to compare groups. This may involve ANOVA or multivariate analysis to better understand the interactions between different treatment variables and their impact on soil structure quality.

Author Response

Please find attached responses from authors for reviewer-3

Round 2

Reviewer 1 Report

Comments and Suggestions for Authors

Thanks for the efforts the authors made for improving the manuscript. While I appreciate the revisions, there remain several issues that need clarification as highlighted during the initial review. These must be addressed before the manuscript can be considered suitable for publication.

Major comments:

1. As previously noted, the detailed parameter settings for each model remain unclear. Given the significant impact of parameter values on prediction performance, it is crucial to specify these parameters in Section 2.4. Although references (Krauss, Do & Huck, 2017; Wu and Kumar, 2009) have been added, they do not provide the requisite detail on parameters.

2. Furthermore, does the parameter setting in references work for your data? I strongly recommend including a parameter setting analysis that demonstrates how to determine optimal parameters for your applications, thereby verifying the applicability of the referenced settings to your data.

Minor comments:

1. Line 268: fig 2a. should be replaced with Fig. 2a..

2. Line 209-291: The figure depicting the SVM decision boundary is not provided as stated.

3. The resolution of Fig. 5 is low, which hampers its utility. Please enhance the image quality to ensure that all details are clearly visible.

4. I am curious about sizes of squares in Fig. 6 are inconsistent.
